# Low efficacy of recombinant SV40 in Ugt1a1$^{-/-}$ mice with severe inherited hyperbilirubinemia

Xiaoxia Shi[1]*, Giulia Bortolussi[2], Lysbeth ten Bloemendaal[1], Suzanne Duijst[1], Andrés F. Muro[2], Piter J. Bosma[1]

1 Amsterdam UMC, Tytgat Institute for Liver and Intestinal Research, AGEM, University of Amsterdam, Amsterdam, The Netherlands, 2 International Centre for Genetic Engineering and Biotechnology, Trieste, Italy

* x.shi@amsterdamumc.nl

## Abstract

In contrast to AAV, Simian Virus 40 (rSV40) not inducing neutralizing antibodies (NAbs) allowing re-treatment seems a promising vector for neonatal treatment of inherited liver disorders. Several studies have reported efficacy of rSV40 in animal models for inherited liver diseases. In all studies the ubiquitous endogenous early promoter controlled transgene expression establishing expression in all transduced tissues. Restricting this expression to the target tissues reduces the risk of immune response to the therapeutic gene. In this study a liver specific rSV40 vector was generated by inserting a hepatocyte specific promoter. This increased the specificity of the expression of hUGT1A1 *in vitro*. However, *in vivo* the efficacy of rSV40 appeared too low to demonstrate tissue specificity while increasing the vector dose was not possible because of toxicity. In contrast to earlier studies, neutralizing antibodies were induced. Overall, the lack of a platform to produce high titered and pure rSV40 particles and the induction of NAbs, renders it a poor candidate for *in vivo* gene therapy.

## Introduction

Crigler-Najjar syndrome (CNs), severe unconjugated hyperbilirubinemia, results from the deficiency of UGT1A1, the enzyme that catalyzes the conjugation of unconjugated bilirubin (UCB) with UDP-glucuronic-acid [1]. The conjugation of the hydrophobic UCB results in water soluble bilirubin glucuronides that can be excreted into bile [2]. If not treated effectively the severe form of CNs is lethal in childhood due to UCB accumulation to levels that cause irreversible brain damage [3]. Both lethality and brain damage can be prevented by intensive phototherapy, a cumbersome treatment that becomes less effective overtime [4, 5]. Most patients therefore do need a liver transplant at some point in their life, a highly invasive treatment with several challenges, like the need for re-transplantation, toxicities and adverse effects associated with long-term immunosuppression [6]. In addition, because of the limited availability of donor organs the patients are at risk to develop brain damage while on the waiting list. Novel therapies are therefore warranted, and recent clinical studies for other liver diseases, like hemophilia B show the potential of liver directed gene therapy [7].

**Data Availability Statement:** All relevant data are within the paper and its Supporting Information files.

**Funding:** Xiaoxia Shi is recipient of a fellowship of the China Scholarship Council (CSC).

**Competing interests:** The authors have declared
that no competing interests exist.

The efficacy of several methods for liver directed gene therapy has been investigated for
CNs in UGT1A1 deficient rodent models. Non-viral gene therapy approaches using lipophilic
nanoparticles do show potential for delivery of mRNA resulting in effective but transient
reduction of serum bilirubin levels in a Ugt1a1 deficient rat [8]. For delivery of DNA the efficiency of this non-viral method still is much lower compared to viral vectors. In addition, several viral vectors, such as rSV40 [9], Adenoviral [10], Lentiviral [11] and Adeno Associated
Viral (AAV) [12] vectors and transposons [13] have been tested in the rat model, while only
AAV vectors were tested in Ugt1a1 deficient mice [14–16]. Of all these viral vectors, AAV vectors are the most advanced and are now tested in clinical trials (NCT03466463 and
NCT03223194).

A major challenge for AAV mediated liver directed gene therapy is the presence of pre-existing Neutralizing Antibodies (NAbs) in a significant percentage of the CN patients [17].
These NAbs will block hepatocyte transduction hampering effective treatment efficacy. The
induction of a high titer of NAbs upon the first AAV administration blocking re-treatment
with this vector, is another important hurdle [18]. Re-treating a patient may be required upon
loss of correction due to liver growth, when treating juveniles, or due to drug or alcohol
induced liver damage, or in patients receiving a sub-optimal vector dose such as those participating in the safety and efficacy studies [19].

In this respect, rSV40 vectors with a low pre-existing immune prevalence seem a promising
option [20, 21]. Also, the reported absence of a cellular response and absence of neutralizing
antibodies upon repeated SV40 administration, render this vector a promising candidate for
liver directed gene therapy [22, 23]. The recently developed novel production cell line ensures
production of batches that are free of large T antigen, a prerequisite for clinical application of
this vector [24].

A potential problem of clinical use of rSV40 vectors is the ubiquitous nature of the endogenous SV40 early promoter. This promoter has been used widely in expression studies and is
suitable to provide expression of a transgene in many different cell types [25, 26]. For *in vivo*
application, this ubiquitous nature is a disadvantage because expression of the therapeutic
UGT1A1 protein, for instance in antigen presenting cells, could increase the risk of an adaptive
immune response. Restricting the expression of UGT1A1 to the hepatocytes will reduce this
risk significantly [12].

In this study an rSV40 vector with a liver specific promotor to drive the expression of a
reporter gene and the therapeutic hUGT1A1 gene was developed and its specificity, efficacy
and immunogenicity was tested *in vitro* and subsequently *in vivo* in a Ugt1a1 deficient mouse
model.

## Material and methods

### Production of viral vectors

To generate liver specific rSV40 vectors a hybrid liver specific promoter (HLP) [27, 28] was
inserted between the endogenous SV40 early promoter and the luciferase or hUGT1A1 cDNA,
using the ClaI and SpeI sites present in Pam310 from AMARNA [24] and rSV-h*UGT1A1* [29].
rSV-*Luc* and rSV-HLP-*Luc* vectors were produced in using co-transfection with a Cre-recombinase expressing plasmid as reported previously [29] or by removal of the bacteria backbone
with Not-I, gel purification and re-ligation [24]. 3 and 6 days after transfection, the medium
was collected and viral vector was concentrated using a spin filter (100 KD, Lot R9NA92290,
Merck Millipore Ltd, Ireland). The rSV-h*UGT1A1* and rSV-HLP-h*UGT1A1* vectors used *in
vitro* and *in vivo* studies were produced in Super-Vero cells by AMARNA Therapeutics (Leiden, The Netherlands) as described [24].

## Transfection, transduction and Western blot analysis

$10^4$ cells per well were seeded in Costar white 96 well plates (ref 3610, corning incorporated Kennebunk, ME, USA). The next day cells were transfected using 40 ng of plasmid and 80 ng of PEI/well or transduced with rSV40 vector (400 vg/cell). Luciferase expression was determined 48 hours after transfection or 2–5 days after transduction by adding 100 μl lysis buffer with substrate (ONE-Glo™ Luciferase, Promega), and after a 15 minutes incubation at room temperature, luminescence was measured in the synergy HT, measurement time was 1 second/well. UGT1A1 expression was determined using western blotting at 2 days after transfection or 2–5 days after transduction. The cells were washed once with PBS and lysed with RIPA (50 mM Tris PH 8.0, 150 mM NaCl, 1% Triton X-100, 0.5% Na-deoxycholate, 0.1% SDS) buffer containing Protease-Inhibitor (1:100 dilution) (Roche, Germany) for 20 minutes. 30 μg of cell lysate was loaded on a 10% Acrylamide gel and blotted to PVDF membrane (semi-dry blotting, 1 hour, 0.05 mA per gel). A monoclonal antibody towards UGT1A1 (1:700 dilution) followed by a goat anti-mouse HRP labeled (Dakoplast, the Netherlands) (1: 5,000) was used to detect UGT1A1 as described previously [12]. An anti-UGT1 rabbit polyclonal antibody (Santa Cruz Biotechnology, Santa Cruz, CA) [28] was used to detect UGT1A1 expression in liver of Ugt1a1 deficient mice treated with rSV40 and for comparison, treated with rAAV8 and in wild type mice.

## Animal study

FVB/NJ background Ugt1a $^{-/-}$ mice were treated with phototherapy (PT) until weaning and housed in a temperature-controlled environment with 12/12-hour light-dark cycle, with a standard diet and water ad libitum in the animal house unit [30]. The Ugt1a$^{-/-}$ mice have a one nt deletion in exon 4 of the Ugt1a locus causing a shift in the reading frame introducing a stop codon immediately after the deletion. Due to the absence of the co-substrate binding and transmembrane domains the truncated enzyme is inactive, as published previously [14]. All animal protocols were approved by the Animal Welfare and Ethics Committee of the International Centre for Genetic Engineering and Biotechnology (ICGEB), Trieste, Italy. At 60 days after birth, the mice were randomized and a single administration of rSV-h*UGT1A1* (n = 7) or rSV-HLP-h*UGT1A1* (n = 8) or a PBS injection control (n = 7) by tail vein injection, 3 mice with dose of $2x10^{11}$ vg/kg, and 4–5 mice with dose of $1.7x10^{12}$ vg/kg, respectively. Intravenous of rSV40 was performed under isoflurane anesthesia. No surgical or other procedures requiring anesthesia were performed. Animals were treated by researchers trained for animal care and monitored daily for general appearance. No signs of stress (lack of appetite, body weight loss, hair loss, stereotypic or aggressive behavior, etc.) or macroscopic alterations of vital functions were observed throughout the experimental phase. Blood was sampled at 1, 3, 5 and 8 weeks after vector administration by facial vein puncture and collected in EDTA-containing tubes. At the time of sacrifice mice were anesthetized with 5% isoflurane, cardiac puncture was performed to obtain blood samples, and sacrificed by cervical dislocation. Plasma was obtained by centrifuging at 400 g for 15 min at room temperature, and stored at -80˚C for further analysis. Total bilirubin determination in plasma was performed following the instructions of the supplier (BQ KITS) as previously described [30]. Organs were directly frozen in liquid nitrogen and stored at -80˚C for further analysis.

## Determination of rSV40 neutralizing antibodies

SV40 NAbs titer in mice plasma were determined using the protocol reported for determination of the NAbs titer towards AAV, using serial dilution of plasma in FCS after complement inactivation by incubation at 56˚C for 30 minutes [31]. Briefly, $10^4$ COS-1 cells/well were

seeded in costar white 96 well plates, in Dulbecco's modified Eagle's medium (DMEM) (Lonza/Westurg) containing 10% fetal calf serum (FCS), 100 U/ml penicillin/streptomycin (Invitrogen) and 20 mM L-Glutamine (Lonza/Westurg) and cultured overnight at 37˚C in 5% $CO_2$. The next day plasma samples were diluted in 7 steps on a half-log scale in FCS. Diluted samples were incubated with an equal volume of $2.5 \times 10^9$ vg/ml rSV-*Luc* at 37˚C for one hour. The maximal transduction control, set to 100%, consisted of rSV-*Luc* incubated with FCS only, the negative control consisted of FCS diluted with medium; both were incubated under the same conditions. After this pre-incubation, 7.5 µl per well of each dilution was added to the COS-1 cells in triplicate. Positive and negative controls were performed in six-fold. The cells were incubated at 37˚C, 5% $CO_2$, and after two days the luciferase expression was determined using ONE-Glo™ Luciferase assay system (Promega), according to manufacturer's protocol.

## Anti-UGT1A1 antibody assay

An indirect ELISA approach was used to detect the anti-UGT1A1 IgG in the mice serum as previously described [32]. Per well of a 96 ELISA plate, 50 µl of recombinant human UGT1A1 protein (Bio Connect) (1 µg/ml) in coating buffer (15 mM $Na_2CO_3$, 35 mM $NaHCO_3$) was added and incubated overnight at 4˚C. The next day the UGT1A1 coating solution was replaced with 100 µl of blocking buffer, 1% gelatine in phosphate-buffered saline (PBS). After 1 hour, the blocking was removed and 50 µl serial dilutions of heparin plasma samples diluted in washing buffer were added to the wells and incubated for 1,5 hours at room temperature. After 3 washings with washing buffer 0.05% Tween-20 in PBS, UGT1A1 binding mouse immunoglobulins were detected with horseradish peroxidase (HRP) conjugated anti-mouse IgG 1:1000 dilution in conjugation buffer containing 4/5 blocking buffer and 1/5 washing buffer followed by o-phenylenediamine (Sigma) conversion.

## Statistical analysis

Statistical significance was determined using GraphPad Prism 8.3.0 (GraphPad Software, La Jolla, CA).

## Results

### Insertion of a liver-specific promoter in rSV40 increases transcriptional activity in liver-derived cells

To restrict the expression provided by an SV40 vector to the liver, a liver specific promoter was inserted between the SV40 early promoter and the luciferase gene generating the rSV-HLP-*Luc* plasmid. In both human hepatoma cell lines, Huh7 and HepG2, transfection of the rSV-HLP-*Luc* plasmid provided a higher luciferase expression than the construct containing only the endogenous SV40 promoter, rSV-*Luc* (Fig 1A and 1B). In non-hepatoma cell lines, COS-1 and HEK293T, the latter was clearly more active (Fig 1C and 1D). This indicates that the liver-specific promoter is functional and significantly increases transcriptional activity in liver-derived cells. To investigate if the tissue-specific promoter remains functional in the viral vector, both plasmids were used to produce recombinant SV40 vector. The hepatoma and non-hepatoma cell lines were transduced using 400 vg/cell of rSV40, and luciferase expression was determined at 2–5 days after transduction. In both hepatoma cell lines transduced with the rSV-HLP-*Luc* vector, the luciferase intensity was significantly higher than in cells transduced with the rSV-*Luc* (Fig 2A and 2B). In contrast, in both non-hepatoma cell lines, the rSV-*Luc* vector provided a higher luciferase intensity (Fig 2C and 2D).

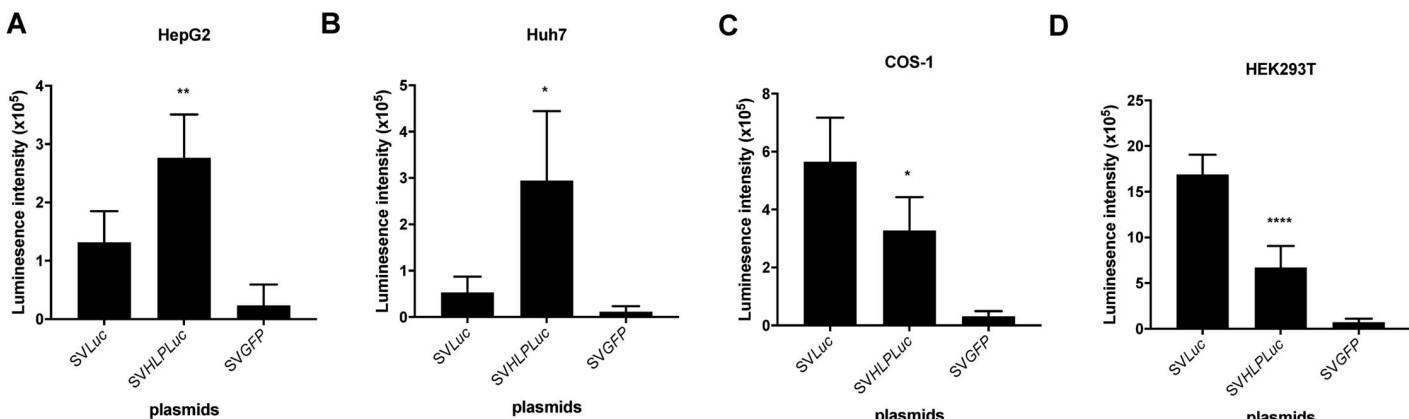

**Fig 1. Insertion of a liver-specific promoter in rSV40-*Luc* plasmid enhances specific expression.** HepG2 (A), Huh7 (B), COS-1 (C), or HEK293T (D) cells were transfected with 40 ng plasmid/$10^4$ cells in a well of a 96 well plate, of rSV-*Luc*, rSV-HLP-*Luc* or rSV-*GFP*, respectively in 6 fold. 48 hours after transfection, luciferase expression was quantified by determining the luminescence intensity. * p <0.05, ** p <0.01, **** p<0.0001. Data represent the mean ± SD. Statistical significance was determined by One-way ANOVA test.

To confirm that insertion of a liver-specific promoter in SV40 gene delivery vector also can improve expression of a therapeutic gene in hepatoma cells, rSV-h*UGT1A1* and rSV-HLP-h*UGT1A1* vectors were generated and used to transfect human hepatoma and non-hepatoma cell lines. The UGT1A1 expression was measured by western blot at 48 hours after transfection and demonstrated that presence of the HLP resulted in a significant increase in HepG2 cells (Fig 3A and 3B). In Huh7 cells, the difference in UGT1A1 expression between both vectors did not reach significance due to large variation. Again in non-hepatoma cell line HEK293T cells, the expression of HLP-driven UGT1A1 was significantly lower compared to the cells transfected with rSV-h*UGT1A1*. This increased transcriptional activity in liver-derived cells induced by presence of the HLP promoter was confirmed upon transduction of these cell lines with the rSV vector generated from these plasmids (Fig 3C and 3D).

### *In vivo*, the poor efficacy of rSV vectors is not improved by the insertion of a liver specific promoter

Upon showing that a liver specific promoter enhances transgene expression in hepatocyte-derived cells, compared to other cell types, the efficacy of both vectors was studied *in vivo*. $2 \times 10^{11}$ vg/kg rSV-h*UGT1A1* or rSV-HLP-h*UGT1A1* were injected in the tail vein of 60-day-old Ugt1a $^{-/-}$ mice. Upon injection the effect on serum bilirubin was monitored overtime. This resulted in at some time points significant, albeit a minor and far from therapeutic, decrease of serum bilirubin in mice injected with rSV-h*UGT1A1* (Fig 4). In the mice treated with the liver specific rSV-HLP-h*UGT1A1*, the serum bilirubin levels were comparable to that in untreated controls, indicating that inserting the hepatocyte specific promoter does not improve *in vivo* efficacy of this vector. A possible explanation for the apparent lower *in vivo* efficacy of the rSV-HLP-h*UGT1A1* compared to the rSV-h*UGT1A1* vector could be a lack of liver tropism of the SV40 vectors, since the latter would result in transduction of many tissues upon injection into a peripheral vein. The bio-distribution of SV40, determined by the ratio between the human UGT1A1 gene carried by the vector and the mouse β-actin gene, showed presence of rSV vector genomes in several tissues. Although in several animals these levels were below detection, the bio-distribution results did not support a specific liver tropism for rSV40 upon injection into a peripheral vein (Table 1). Subsequently, in a higher dose of $1.7 \times 10^{12}$ vg/kg rSV-h*UGT1A1* and of rSV-HLP-h*UGT1A1* was tested (Fig 4). At the time of injection, the

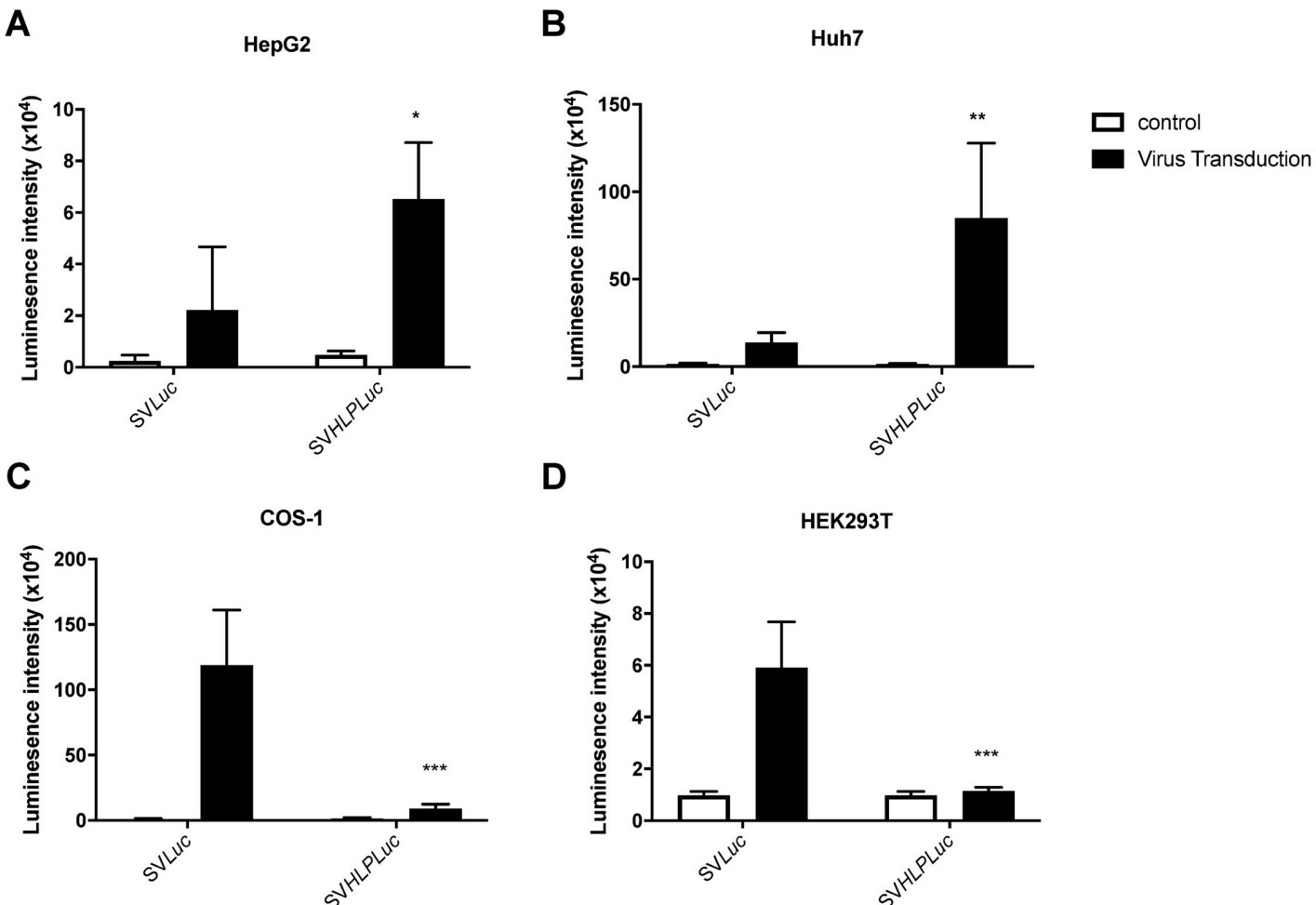

**Fig 2. Insertion of a liver specific promoter in rSV40 vectors enhances specific expression of reporter gene.** 400 vg/cell of rSV40-*Luc* or rSV40-HLP-*Luc* were added to $10^4$ cells of HepG2 (A), Huh7 (B), COS-1 (C), or HEK293T (D) in a well of a 96 well plate (n = 3). After 2–5 days the luciferase expression was quantified by measuring the luminescence using the synergy HT machine. * p <0.05, ** p <0.01, *** p<0.001. Data represent the mean ± SD. Statistical significance was determined by Two-way ANOVA test.

serum bilirubin levels in this experiment were lower due to the natural variation occurring in this model. Administration of the higher vector dose did not result in therapeutic correction albeit that after one week the levels in the mice receiving rSV-HLP-h*UGT1A1* did have lower serum level. The hUGT1A1 expression in liver was determined by western blot showing signals in mice treated with rSV-h*UGT1A1* or rSV-HLP-h*UGT1A1* comparable to that in PBS treated Ugt1a1 deficient mice (Fig 4D). In contrast administration of a 10 fold lower dose of AAV8 vector, containing hUGT1A1 behind the same liver specific promoter (AAV-L), did show presence of hUGT1A1 while treatment with a comparable dose of this vector (AAV-H), showed a very prominent presence of UGT1A1. Two of the mice injected with rSV-h*UGT1A1* died within 24 hours after the injection.

## Intravenous injection of rSV40 does induce a neutralizing antibody response

Absence of a humoral response towards rSV40 would allow repeated administrations of this vector. Retreatment in case of loss of efficacy or upon sub-optimal dosing would be a major

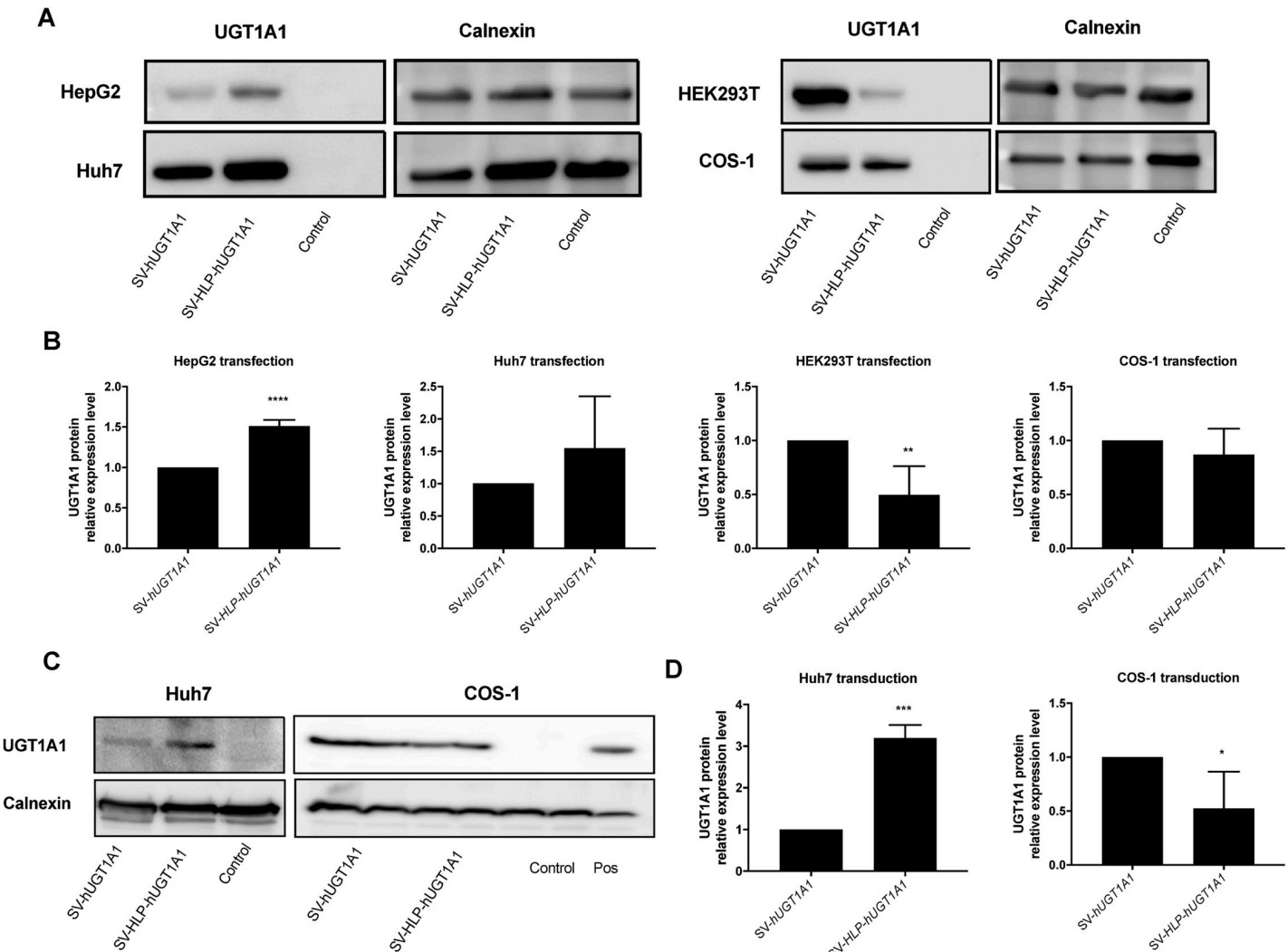

**Fig 3. Insertion of a liver specific promoter in rSV40 provides specific expression of a therapeutic gene.** (A) $10^5$ cells of HepG2, Huh7, COS-1, or HEK293T cells were transfected with 1 μg of rSV-h*UGT1A1* or, rSV-HLP-h*UGT1A*. 48 hours later, UGT1A1 expression was determined by western blotting and quantified normalizing to calnexin, a membrane protein (B). (C) $10^5$ cells of Huh7 and COS-1 cells were transduced with 400 vg/cell of rSV-h*UGT1A or* rSV-HLP-h*UGT1A1*. After 2–5 days the cells were lysed and UGT1A1 expression was detected by western blotting and quantified by western blotting normalized to calnexin. All experiments were repeated 3–5 times. * p <0.05, ** p <0.01, *** p<0.001, **** p<0.0001. Data represent the mean ± SD. Statistical significance was determined by two-tailed Student t test.

advance for rSV40 compared to AAV and Adenoviral vectors. To determine the presence of neutralizing antibodies COS-1 cells were transduced with rSV-*Luc* vector pre-incubated with serum from naïve mice or mice injected with rSV40. After 48 hours transduction efficiency was determined by measuring the luciferase expression in COS-1 cells. Pre-incubation with serum from mice injected with rSV40 did impair luciferase expression, indicating that intravenous administration of rSV40 does induce neutralizing antibodies towards this vector *in vivo*. Comparing to the mice before administration, the anti-SV40 neutralizing antibody titer increased in rSV40 treated mice but not in PBS treated ones (Fig 5). No humoral response towards the encoded hUGT1A1 was detectable irrespective of the presence of the hepatocyte-specific promoter restricting hUGT1A1 expression to the liver or to several tissues when its expression was controlled by the ubiquitous SV40 early promoter (S1 Fig).

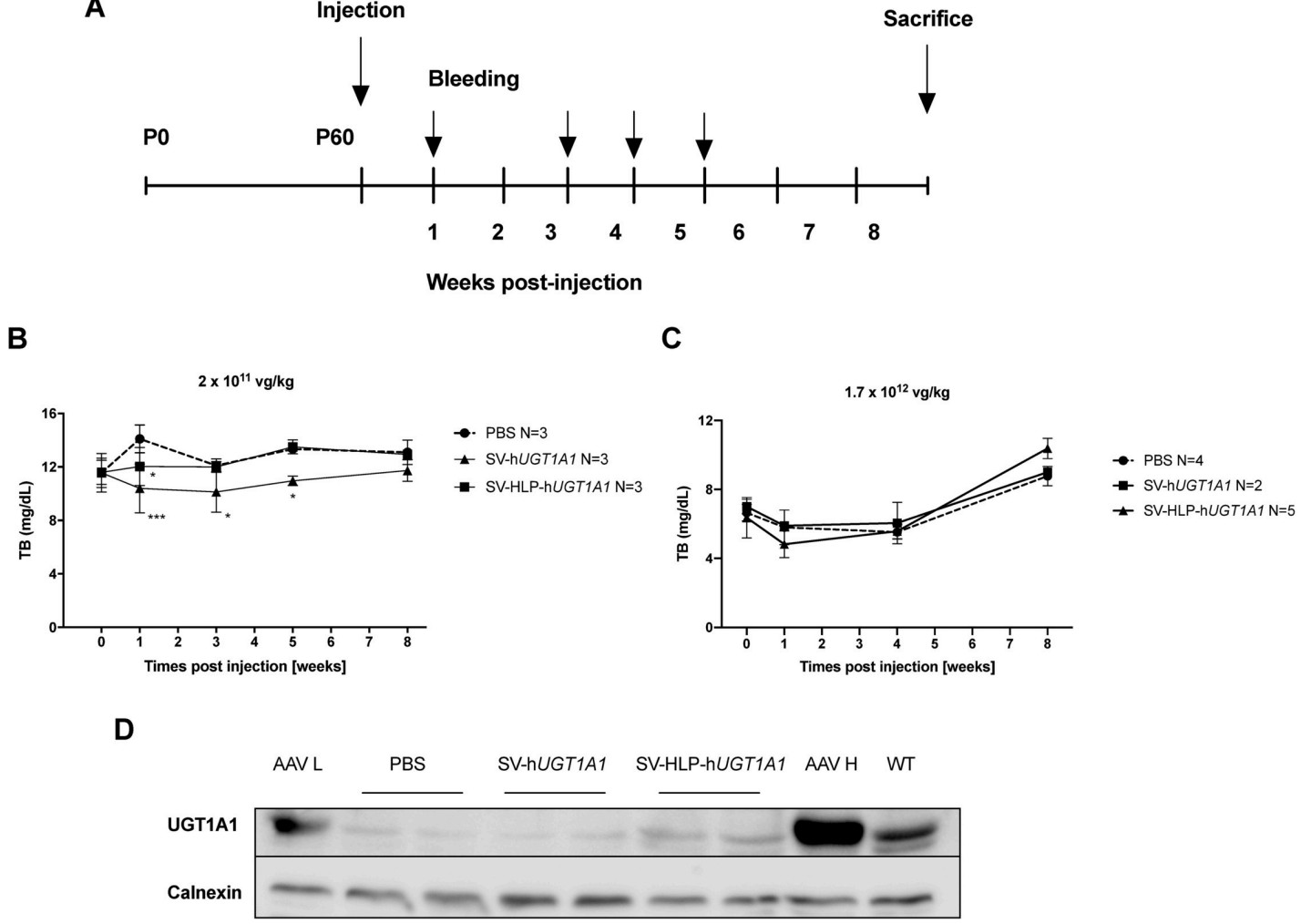

**Fig 4. Insertion of liver specific promoter does not improve the low efficacy of rSV40 vectors *in vivo*.** (A) 60-day-old Ugt1a $^{-/-}$ mice were treated with rSV-h*UGT1A1*, rSV-HLP-h*UGT1A1*, or PBS via tail vein injection. (B-C) Total serum bilirubin (mg/dL) was monitored overtime until week 8 after injection. (D) Presence of UGT1A1 in Ugt1a1 deficient mice treated with $1x10^{11}$ vg/kg of AAV8-h*UGT1A1* (AAV L), PBS, $1.7x10^{12}$ vg/kg of rSV-hUGT1A1, $1.7x10^{12}$ vg/kg of rSV-HLP-h*UGT1A1*, or $1x10^{12}$ vg/kg of AAV8-h*UGT1A1* (AAV H), and in a non-treated wild type mouse (WT). Calnexin was used as loading control on the same membrane. Data represent the mean ± SD. Statistical significance was determined by Two-way ANOVA with Dunnett's multiple comparison test.

## Discussion

The data reported in this study show that insertion of a liver-specific promoter results in more specific expression of transgenes encoded by rSV40 vectors *in vitro*. The insertion of a

**Table 1. Bio-distribution of rSV vector genomes.**

| Treatment | | Liver | Spleen | heart | colon | jejunum | muscle | brain |
|---|---|---|---|---|---|---|---|---|
| SV-h*UGT1A1* | mouse 1 | UD | 4.4E-03 | UD | 3.4E-03 | UD | UD | UD |
| SV-h*UGT1A1* | mouse 2 | UD | UD | UD | 4.0E-03 | UD | 3.4E-03 | 8.3E-03 |
| SV-h*UGT1A1* | mouse 3 | UD | 3.9E-03 | 7.0E-03 | 3.3E-03 | UD | 2.2E-03 | UD |
| SV-HLP-h*UGT1A1* | mouse 4 | 3.3E-03 | UD | UD | 4.2E-03 | 4.4E-03 | UD | UD |
| SV-HLP-h*UGT1A1* | mouse 5 | 3.7E-03 | 4.5E-03 | UD | UD | UD | 4.3E-03 | 6.6E-03 |
| SV-HLP-h*UGT1A1* | mouse 6 | UD | UD | UD | 4.9E-03 | 2.9E-03 | 4.1E-03 | UD |

Note: the data represents the value of 2x(UGT1A1/B-actin), UD represents undetectable.

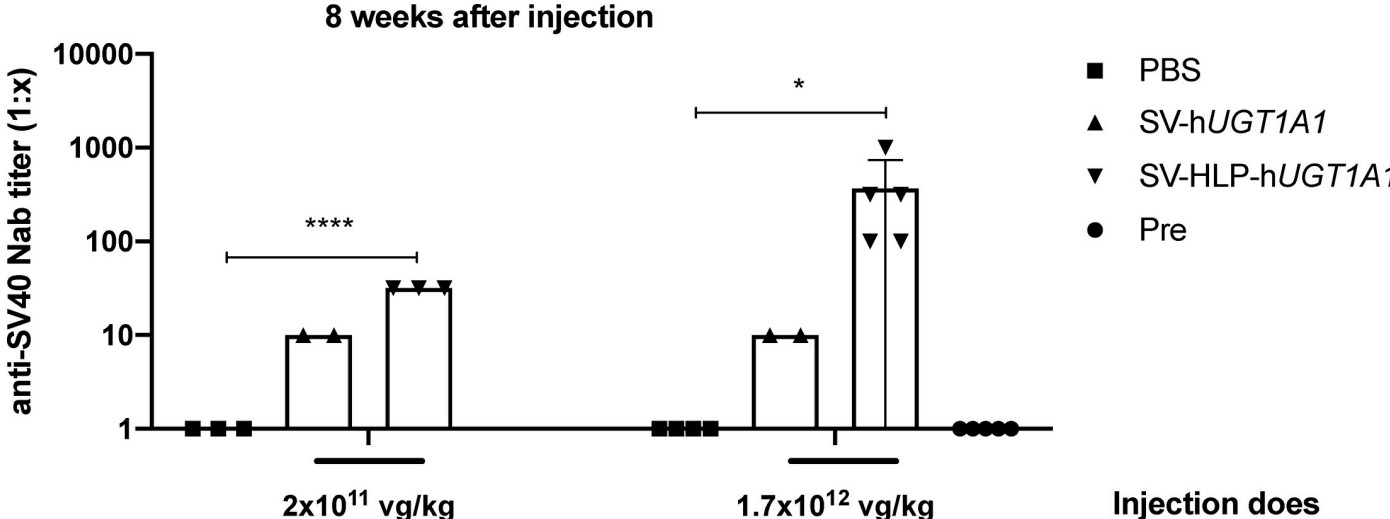

**Fig 5. Intravenous injection does induce neutralizing antibodies towards rSV40.** 60-day-old Ugt1a $^{-/-}$ mice were treated with a lower dose of $2 \times 10^{11}$ vg/kg of rSV-hUGT1A1 vector (n = 3), or rSV-HLP-hUGT1A1 (n = 3), and a higher dose of $1.7 \times 10^{12}$ vg/kg of rSV-hUGT1A1 vector (n = 4), or rSV-HLP-hUGT1A1 (n = 5) via tail vein injection. Serum samples at 8 weeks after vector administration were tested for presence of NAbs towards rSV40 vectors using an *in vitro* transduction assay. NAbs titers are expressed as the highest serum dilution that inhibited AAV transduction by $\geq 50\%$ compared with the control without serum. Pre represents before vector injection. Data represent the mean ± SD. Statistical significance was determined by Ordinary one-way ANOVA with Dunnett's multiple comparison test.

hepatocyte specific promoter did not improve the efficacy of the rSV40 vector *in vivo*. A dose of $1.7 \times 10^{12}$ vg/kg of these vectors did not result in a relevant reduction of serum bilirubin levels. Intravenous administration of both rSV40 vectors did result in the induction of neutralizing antibodies in all animals, rendering effective retreatment with this vector most unlikely.

*In vivo* gene therapy using rSV40 vector has been reported for unconjugated hyperbilirubinemia [9]. A replication-deficient rSV40 mediated a long-term amelioration of jaundice in Gunn rats. In that study, serum bilirubin levels were lowered by 35% upon a single injection, while in this study the reduction using a similar rSV40 vector in a UGT1A1 deficient mouse model is only about 10%. The different administration route, peripheral vein versus portal vein, may explain the difference in correction since upon peripheral vein injection no clear liver tropism for rSV40 was seen. The bio-distribution showed transduction of many tissues by rSV40 albeit all at a low level (Table 1). The lack of a liver tropism can also explain the better performance of the non-hepatocyte specific rSV40 vector since UGT1A1 expression in non-hepatic tissues also provides effective conjugation of UCB [15, 33, 34]. Administration into a peripheral vein of AAV8, a vector that does display a clear liver tropism, provides a much more efficient correction. A dose of $1 \times 10^{11}$ vg/kg results in therapeutic correction of serum bilirubin levels [28] and in contrast to both rSV40 vectors, in clearly detectable levels of UGT1A1 in the liver (Fig 4D). In the Gunn rat, at least three repeated administrations of $3 \times 10^{9}$ I.U./ rat of 200–300 g, resulting in a total dose of $3 \times 10^{10}$ I.U./kg of rSV40 were needed to provide a reduction of serum bilirubin by 70% [9]. In this animal model, AAV does provide therapeutic correction upon peripheral administration [28]. A dose of $5 \times 10^{12}$ AAV8 vg/kg comparable to $2 \times 10^{8}$ Huh7 transducing AAV8 units, results in sustained and complete normalization of serum bilirubin [35]. These data indicate that the liver transduction efficacy of AAV vectors is higher than that of rSV40 vectors. This difference in liver transduction efficacy between AAV and SV40 vectors in rats has also been observed by others [36]. In that study a dose of $3.4 \times 10^{9}$ vg of AAV1-IGF1 provided a comparable expression of IGF-1 as $1 \times 10^{11}$ vg of rSV40-*IGF-1* per rat, which is a 30 fold higher dose. Although, comparing doses of different viral vectors is

complicated by different titration methods, the higher efficacy seen with AAV vector renders that vector more suitable for *in vivo* gene therapy for inherited liver disorders. Another major hurdle for rSV40 vectors is the absence of a production and purification protocol to generate sufficiently high titered vector batches, which rendered testing the therapeutic efficacy of higher rSV40 vector doses *in vivo* is not possible because of toxicity. The cause of the toxicity seen in two out of four mice upon injection of the high dose of rSV-h*UGT1A1* suggest the presence of impurities in our vector batch.

In the Gunn rat, repeated injections with rSV-h*UGT1A1* did not inhibited liver transduction of rSV40-*HBs* antigen [9]. This indicated that rSV40 did not induce neutralizing antibodies. Others also reported the absence of NAbs upon injection of rSV40 [22]. Absence of a NAbs response would be a major advantage for rSV40 compared to AAV since it renders effective retreatment feasible upon loss of efficacy. Using an *in vitro* NAbs assay, comparable to that used to test for a response towards AAV, we observed the induction of an immune response that prevented transduction *in vitro*. At 8 weeks after rSV40 administration, serum from animals treated with rSV40 completely prevented the transduction of COS-1 cells. The observation that in Gunn rats re-administration was effective may be explained by the high dose of rSV40 administered locally to the liver via the portal vein. Presence of NAbs against AAV can, at least in part, be overcome by a high dose of vector or by co-administration of empty particles as a decoy [37]. Upon portal delivery the amount of NAbs present locally may not be sufficient to block the liver transduction completely. Although we cannot exclude that the presence of impurities in the vector batches used may have had an adjuvant effect resulting in a more prominent immune response seen in this study. However, the production system used in these studies is very similar to that used in those Gunn rat studies rendering such an explanation less probable.

Presence of pre-existing immunity towards viral gene therapy vectors is a challenge for AAV mediated *in vivo* gene therapy for liver disorders. The presence of NAbs renders about 30% of the Crigler-Najjar patients screened not eligible for the ongoing trial [17]. In this respect, the lower pre-existing immunity towards rSV40 is an advantage. In the general population seroconversion towards SV40 immunity is observed at a young age [38]. Most studies report about 10 to 25% of all subject have antibodies to this vector. The percentage of the population that has neutralizing antibodies towards SV40 is much lower [21, 39]. Although this shows that pre-existing immunity towards rSV40 is clearly lower than that reported for AAV, these studies do report presence of neutralizing antibodies in the normal healthy population. This presence supports our finding that injection of rSV40 vectors does induce neutralizing antibodies, which argues against the possibility of effective re-administration indicating this remains a hurdle for rSV40 vector also.

## Conclusion

Our data indicate that a platform to generate high titered pure rSV40 vector batches is needed to support the potential use of rSV40 as an additional gene therapy vector to treat inherited liver diseases. The current vector batches have, compared to for instance AAV, a much lower transduction efficacy and lack the previously reported immune privilege could not be reproduced. Presently, the only advantage of rSV40 is the lower pre-existing immunity towards this vector in the general population.

## Supporting information

**S1 Fig. Absence of antibodies towards hUGT1A1 in serum of mice treated with rSV h*UGT1A1* or rSV-HLP-h*UGT1A1*.** 60-day-old Ugt1a$^{-/-}$ mice were treated with 2x1011 vg/kg

of rSV-h*UGT1A1* vector (n = 3), or rSV-HLP-h*UGT1A1* (n = 3) via tail vein injection. At 8 weeks after vector administration the level of anti-hUGT1A1 IgG in serum was determined. Data represent the mean ± SD.
(PDF)

**S1 Raw images.**
(PDF)

## Acknowledgments

We thank AMARNA Therapeutics for the production of SV-h*UGT1A1* and SV-HLP-h*UGT1A1* viral vector batches used in vivo.

## Author Contributions

**Conceptualization:** Piter J. Bosma.

**Data curation:** Xiaoxia Shi, Giulia Bortolussi, Lysbeth ten Bloemendaal.

**Formal analysis:** Xiaoxia Shi.

**Methodology:** Giulia Bortolussi, Suzanne Duijst.

**Project administration:** Piter J. Bosma.

**Resources:** Giulia Bortolussi, Andrés F. Muro.

**Supervision:** Piter J. Bosma.

**Writing – original draft:** Xiaoxia Shi, Piter J. Bosma.

**Writing – review & editing:** Xiaoxia Shi, Giulia Bortolussi, Lysbeth ten Bloemendaal, Suzanne Duijst, Andrés F. Muro, Piter J. Bosma.

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
