## [Decision Letter · Decision Letter 0]

22 Dec 2020

PONE-D-20-33243

Lack of purification method to produce pure high tittered batches, renders rSV40 a poor candidate for in vivo liver directed gene therapy

PLOS ONE

Dear Dr. Shi,

Thank you for submitting your manuscript to PLOS ONE. After careful consideration, we feel that it has merit but does not fully meet PLOS ONE’s publication criteria as it currently stands. Therefore, we invite you to submit a revised version of the manuscript that addresses the points raised during the review process.

We look forward to receiving your revised manuscript.

Kind regards,

Chen Ling, Ph.D.

Academic Editor

PLOS ONE

Journal Requirements:

2. Thank you for including your ethics statement:  "Mice were maintained in the International Centre for Genetic Engineering and Biotechnology animal house unit. All animal studies were approved by the local ethical committee. ".   

Please amend your current ethics statement to include the full name of the ethics committee that approved your specific study.

For additional information about PLOS ONE submissions requirements for ethics oversight of animal work, please refer to http://journals.plos.org/plosone/s/submission-guidelines#loc-animal-research  

3. In your Methods section, please provide additional information on the animal research and ensure you have included details on : (1) methods of sacrifice (2) methods of anesthesia and/or analgesia, and (2) efforts to alleviate suffering.

"Xiaoxia Shi is recipient of a fellowship of the China Scholarship Council (CSC)."

Reviewers' comments:

Reviewer's Responses to Questions

**Comments to the Author**

1. Is the manuscript technically sound, and do the data support the conclusions?

Reviewer #1: Yes

Reviewer #2: Yes

2. Has the statistical analysis been performed appropriately and rigorously? 

Reviewer #1: No

Reviewer #2: Yes

3. Have the authors made all data underlying the findings in their manuscript fully available?

Reviewer #1: Yes

Reviewer #2: Yes

4. Is the manuscript presented in an intelligible fashion and written in standard English?

Reviewer #1: No

Reviewer #2: Yes

5. Review Comments to the Author

Reviewer #1: This paper describes simian virus 40 (rSV40) is difficult to serve as candidate vector for in vivo gene therapy, due inefficient production and induction of immune neutralization. The study provides a novel sight that rSV40 triggered neutralizing antibody production, inconsistent with previous reports. Most of results are solid and convincing. However, some concerns and more experiments are need to further verify.

1. In vitro, authors used the human liver-derived cell lines (Huh7 and HepG2) to test the activation of hybrid liver specific promoter, they should use mouse liver-derived cell lines to test the promoter activity as well.

2. Authors are necessary to detect the expression of hUGT1A1 in mRNA and protein levels in liver from mice injected by rSV40 vectors.

3. In figure 4 and 5, statistical analysis methods should be included in figure legends.

4. There is no data description the “lack of purification method to produce pure high tittered batches” in title, hence the title needs to be amended.

5. Authors needs to present some data about the Ugt1a gene being knocked out in mice.

6. English/grammar should be improved for clarity, e.g. in 173-174 line, wass should be corrected to was.

Reviewer #2: This manuscript basically reported negative results. The reviewer appreciated the hard work and the authors’ scientific integrity. It is not clear whether the same viral vectors were used for in vitro and in vivo experiments. The expression of transgene in the liver should be detected by Western blot.

Minors,

Line 44, why there is only one sub-title for Introduction?

Line 204, rSV40 should be rSV.

6. PLOS authors have the option to publish the peer review history of their article (what does this mean?). If published, this will include your full peer review and any attached files.

Reviewer #1: No

Reviewer #2: **Yes: **Chen Ling

---

## [Author Response · Author response to Decision Letter 0]

1 Feb 2021

Dr. P. J. Bosma

Amsterdam University Medical Centers

Tytgat Institute for Liver Research

Meibergdreef 69

1105 BK Amsterdam

The Netherlands

Dr. Joerg Heber

Editor-in-Chief

PLOS ONE

February 1st 2021

Dear dr. Joerg Heber,

We are grateful for your potential interest in our manuscript entitled manuscript “Lack of purification method to produce pure high tittered batches, renders rSV40 a poor candidate for in vivo liver directed gene therapy (ID: PONE-D-20-33243)”. We thank the reviewers for their comments and suggestions to improve our manuscript. A manuscript with the tracked changes in red and a clean version are submitted. Please find our point-by-point response below.

Yours sincerely,

Xiaoxia Shi

Dr. Piter Bosma

Journal Requirements:

2. Thank you for including your ethics statement:  "Mice were maintained in the International Centre for Genetic Engineering and Biotechnology animal house unit. All animal studies were approved by the local ethical committee. ".   

Please amend your current ethics statement to include the full name of the ethics committee that approved your specific study.

For additional information about PLOS ONE submissions requirements for ethics oversight of animal work, please refer to http://journals.plos.org/plosone/s/submission-guidelines#loc-animal-research  

Response: we have amended the ethics statement according to the requirement in the manuscript.

Line 122-124: All animal protocols were approved by the Animal Welfare and Ethics Committee of the International Centre for Genetic Engineering and Biotechnology (ICGEB), Trieste, Italy. 

3. In your Methods section, please provide additional information on the animal research and ensure you have included details on: (1) methods of sacrifice (2) methods of anesthesia and/or analgesia, and (2) efforts to alleviate suffering.

Response: According to the requirements, we have added the details in the manuscript.

Line 127-134: Intravenous of rSV40 was performed under isoflurane anesthesia. No surgical or other procedures requiring anesthesia were performed. Animals were treated by researchers trained for animal care and monitored daily for general appearance. No signs of stress (lack of appetite, body weight loss, hair loss, stereotypic or aggressive behavior, etc.) or macroscopic alterations of vital functions were observed throughout the experimental phase. Blood was sampled at 1, 3, 5 and 8 weeks after vector administration by facial vein puncture and collected in EDTA-containing tubes. At the time of sacrifice mice were anesthetized with 5% isoflurane, cardiac puncture was performed to obtain blood samples, and sacrificed by cervical dislocation. 

Response: We have provided the original underlying images for all blot reported in our manuscript with the name of S1_raw_images.

"Xiaoxia Shi is recipient of a fellowship of the China Scholarship Council (CSC)."

Response: We have deleted the sentence ‘Xiaoxia Shi is recipient of a fellowship of the China Scholarship Council (CSC)’ in the Acknowledgments section and want to include this in the Funding Statement.

Response: we have added the results as supplemental figure with the name of S1 Fig.

 Reviewers' comments: 

Reviewer #1: This paper describes simian virus 40 (rSV40) is difficult to serve as candidate vector for in vivo gene therapy, due inefficient production and induction of immune neutralization. The study provides a novel sight that rSV40 triggered neutralizing antibody production, inconsistent with previous reports. Most of results are solid and convincing. However, some concerns and more experiments are need to further verify.  1. In vitro, authors used the human liver-derived cell lines (Huh7 and HepG2) to test the activation of hybrid liver specific promoter, they should use mouse liver-derived cell lines to test the promoter activity as well. 

Response: We agree that in view of the poor efficacy in vivo, showing that the promoter used is active in mouse hepatocytes is relevant. Although we have not tested this promoter in mouse hepatoma cells in vitro, we have used it activity of this promoter has been demonstrated in vivo in mouse liver. In AAV vectors this promoter provides the expression of UGT1A and human UGT1A1 in mouse liver upon systemic injection. Since this is indeed relevant we inserted the references to these previously published studies. The reference 27 and 28.

 2. Authors are necessary to detect the expression of hUGT1A1 in mRNA and protein levels in liver from mice injected by rSV40 vectors.

 Response: Showing mRNA expression is indeed a final prove of the expression from these constructs. We have determined mRNA expression using qPCR in the liver of the treated animals. In contrast to our studies using AAV, the mRNA levels in mice treated with the rSV40 vector were very low. Although a signal was detected in some of the treated mice, the levels were too low for reliable quantification using this very sensitive method, again confirming that rSV40 vectors appear very in-efficient for liver transduction in vivo.

 3. In figure 4 and 5, statistical analysis methods should be included in figure legends.

Response: We do apologize for this omission and thank the reviewer, and have included the statistical analysis methods in figure 4 and 5. 

 4. There is no data description the “lack of purification method to produce pure high tittered batches” in title, hence the title needs to be amended. 

Response: we thank the reviewer for this comment. As mentioned a good production system for rSV40 has been developed using the newly developed super Vero cell line. The generation of high titered vector needed for in vivo experiments depends on highspeed centrifugation. No methods have been developed to ensure the removal of cellular debris. Presence of this, consisting of cell membrane and DNA, can be toxic upon administration to the systemic circulation, explaining the toxicity seen in mice when increasing the dose. We do agree with the reviewer that we did not investigate the toxicity nor did we show presence of cell debris. Therefore we do agree with changing the title: Low efficacy of rSV40 in Ugt1a1-/- mice with severe inherited hyperbilirubinemia. 

 5. Authors needs to present some data about the Ugt1a gene being knocked out in mice.

Response: We thank the reviewer for this comment and have added additional describing the generation of this model and the data obtained in this model in our previous study (Bortolussi et al, 2012, FASEB J.)

Line 119-122: The Ugt1a-/- mice have a one nt deletion in exon 4 of the Ugt1a locus causing a shift in the reading frame introducing a stop codon immediately after the deletion. Due to the absence of the co-substrate binding and transmembrane domains the truncated enzyme is inactive, as published previously (14).

 6. English/grammar should be improved for clarity, e.g. in 173-174 line, wass should be corrected to was.

Response: we thank the reviewer for helping us and have corrected this mistake.

Reviewer #2: This manuscript basically reported negative results. The reviewer appreciated the hard work and the authors’ scientific integrity. It is not clear whether the same viral vectors were used for in vitro and in vivo experiments. The expression of transgene in the liver should be detected by Western blot.

Response: we thank the reviewer for your recognition of our work. The viral vectors used in vitro and in vivo were identical. Both rSV-hUGT1A1 and rSV-HLP-hUGT1A1 were produced by AMARNA Therapeutics in Leiden, the Netherlands.

By treating the mutant mice with SV40 vectors, the expression of the transgene was too low to be detected by WB. We have seen in other gene therapy experiments using AAV vectors, that if the efficacy of the treatment is not enough to reduce plasma bilirubin levels below 3-4 mg/dL, the transgenic protein cannot be detected by WB analysis.

 Minors, Line 44, why there is only one sub-title for Introduction?

Response: We agree with the reviewer that this one sub-title is strange and have removed it. 

 Line 204, rSV40 should be rSV.

Response: we thank the reviewer’s point out that, and we change rSV40 to rSV in the manuscript.

---

## [Decision Letter · Decision Letter 1]

23 Feb 2021

PONE-D-20-33243R1

Low efficacy of recombinant SV40 in Ugt1a1-/- mice with severe inherited hyperbilirubinemia

PLOS ONE

Dear Dr. Shi,

Thank you for submitting your manuscript to PLOS ONE. After careful consideration, we feel that it has merit but does not fully meet PLOS ONE’s publication criteria as it currently stands. Therefore, we invite you to submit a revised version of the manuscript that addresses the points raised during the review process.

We look forward to receiving your revised manuscript.

Kind regards,

Chen Ling, Ph.D.

Academic Editor

PLOS ONE

Reviewers' comments:

Reviewer's Responses to Questions

**Comments to the Author**

1. If the authors have adequately addressed your comments raised in a previous round of review and you feel that this manuscript is now acceptable for publication, you may indicate that here to bypass the “Comments to the Author” section, enter your conflict of interest statement in the “Confidential to Editor” section, and submit your "Accept" recommendation.

Reviewer #1: All comments have been addressed

Reviewer #2: (No Response)

2. Is the manuscript technically sound, and do the data support the conclusions?

Reviewer #1: Yes

Reviewer #2: Yes

3. Has the statistical analysis been performed appropriately and rigorously? 

Reviewer #1: Yes

Reviewer #2: Yes

4. Have the authors made all data underlying the findings in their manuscript fully available?

Reviewer #1: Yes

Reviewer #2: Yes

5. Is the manuscript presented in an intelligible fashion and written in standard English?

Reviewer #1: Yes

Reviewer #2: Yes

6. Review Comments to the Author

Reviewer #1: Authors have complemented the revision. I do not any other questions. If it is allowed，i hope these authors to provide a higher aesthetic picture！

Reviewer #2: The reviewer is disappointed that the authors decline to perform additional experiments. The reviewer understand that this manuscript basically reported negative results. But to make the story complete, some missing data should be provided. Only citing an almost ten-year old reference (Bortolussi et al, 2012, FASEB J.) is not enough. The reviewer recommends a Western Blot assay showing Ugt1a1 expression in the mice liver. The groups should include WT mice, Ugt1a1-/- mice, rSV40 treated mice, and probably rAAV treated mice from the previous reports. The authors should retain some tissues from the previous research. It is very hard to understand why the authors decline to perform a simple WB assay. The ratio of the rSV40- and rAAV-mediated transgene expression is very important for the reader to make a conclusion.

7. PLOS authors have the option to publish the peer review history of their article (what does this mean?). If published, this will include your full peer review and any attached files.

Reviewer #1: No

Reviewer #2: No

---

## [Author Response · Author response to Decision Letter 1]

7 Apr 2021

Reviewers' comments: 

Reviewer #1: Authors have complemented the revision. I do not any other questions. If it is allowed，i hope these authors to provide a higher aesthetic picture！

Response: We thank the review’s suggestion, next time we will use a more professional tool to make figures and pictures. 

Reviewer #2: The reviewer is disappointed that the authors decline to perform additional experiments. The reviewer understand that this manuscript basically reported negative results. But to make the story complete, some missing data should be provided. Only citing an almost ten-year old reference (Bortolussi et al, 2012, FASEB J.) is not enough. The reviewer recommends a Western Blot assay showing Ugt1a1 expression in the mice liver. The groups should include WT mice, Ugt1a1-/- mice, rSV40 treated mice, and probably rAAV treated mice from the previous reports. The authors should retain some tissues from the previous research. It is very hard to understand why the authors decline to perform a simple WB assay. The ratio of the rSV40- and rAAV-mediated transgene expression is very important for the reader to make a conclusion.

Response: Since using a very sensitive qPCR assay we did show the expression of hUGT1A1 mRNA in the rSV40 treated animals is close to background levels the presence of UGT1A1 protein in liver is expected to be undetectable. We do understand the point raised by the reviewer that a western will further strengthen our message that liver transduction by rSV40 vectors is very inefficient and have therefore performed this experiment. Using an anti-UGT1 rabbit polyclonal antibody (Santa Cruz Biotechnology, Santa Cruz, CA) to detect UGT1A1 expression in liver of Ugt1a1 deficient mice treated with rSV40 and for comparison, treated with rAAV8 and in wild type mice. This western has been added to the paper (Fig 4D) and indeed illustrates the major difference in efficacy of these two vectors in a very clear way. We therefore thank this reviewer for being persistent at this point. 

The results of this western are presented as follows:

Line 223-228: The hUGT1A1 expression in liver was determined by western blot showing signals in mice treated with rSV-hUGT1A1 or rSV-HLP-hUGT1A1 comparable to that in PBS treated Ugt1a1 deficient mice (Fig 4D). In contrast administration of a 10 fold lower dose of AAV8 vector, containing hUGT1A1 behind the same liver specific promoter (AAV-L), did show presence of hUGT1A1 while treatment with a comparable dose of this vector (AAV-H), showed a very prominent presence of UGT1A1.

This result is also mentioned in the discussion as follows: 

Line 268-270: A dose of 1x1011 vg/kg results in therapeutic correction of serum bilirubin levels (28) and in contrast to both rSV40 vectors, in clearly detectable levels of UGT1A1 in the liver (Fig 4D).

The legend of Fig 4D reads as follows:

Line 475-478: (D) Presence of UGT1A1 in Ugt1a1 deficient mice treated with 1x1011 vg/kg of AAV8-hUGT1A1 (AAV L), PBS, 1.7x1012 vg/kg of rSV-hUGT1A1, 1.7x1012 vg/kg of rSV-HLP-hUGT1A1, or 1x1012 vg/kg of AAV8-hUGT1A1 (AAV H), and in a non-treated wild type mouse (WT). Calnexin was used as loading control on the same membrane.

---

## [Editor Report · Decision Letter 2]

12 Apr 2021

Low efficacy of recombinant SV40 in Ugt1a1-/- mice with severe inherited hyperbilirubinemia

PONE-D-20-33243R2

Dear Dr. Shi,

We’re pleased to inform you that your manuscript has been judged scientifically suitable for publication and will be formally accepted for publication once it meets all outstanding technical requirements.

Kind regards,

Chen Ling, Ph.D.

Academic Editor

PLOS ONE
---

## [Editor Report · Acceptance letter]

15 Apr 2021

PONE-D-20-33243R2 

Low efficacy of recombinant SV40 in Ugt1a1^-/-^ mice with severe inherited hyperbilirubinemia 

Dear Dr. Shi:

I'm pleased to inform you that your manuscript has been deemed suitable for publication in PLOS ONE. Congratulations! Your manuscript is now with our production department. 

Kind regards, 

on behalf of

Dr. Chen Ling 

Academic Editor

PLOS ONE